# Electrochromic Properties of Lithium-Doped Tungsten Oxide Prepared by Electron Beam Evaporation

**Jui-Yang Chang [1], Ying-Chung Chen [1], Chih-Ming Wang [2,*], Wen-Nan Wang [1], Chih-Yu Wen [1] and Jyun-Min Lin [1]**

[1]   Department of Electrical Engineering, National Sun Yat-sen University, Kaohsiung 804, Taiwan;
     twm0929137151@gmail.com (J.-Y.C.); ycc@mail.ee.nsysu.edu.tw (Y.-C.C.);
     hanne820219@yahoo.com.tw (W.-N.W.); kood1555@gmail.com (C.-Y.W.);
     d983010002@student.nsysu.edu.tw (J.-M.L.)
[2]   Department of Electrical Engineering, Cheng Shiu University, Kaohsiung 833, Taiwan
*   Correspondence: cmwang@gcloud.csu.edu.tw; Tel.: +886-7-7358800 (ext. 3410)

**Abstract:** In this study, $x$Li$_2$O-$(1-x)$WO$_3$ powders were mixed with WO$_3$ and Li$_2$O and pressed into target pellets to fabricate electrochromic films on indium tin oxide (ITO) glasses prepared by electron beam evaporation under the parameters of room temperature, and thicknesses of about 530 nm. It was expected that the amount of charge stored in the electrochromic devices (ECDs) could be enhanced by using the doping method in the cathode materials. The experimental results show that as the composition of Li$_{0.18}$W$_{0.82}$O$_{2.6}$ powder was formed, the optimal characteristics of ECD can be obtained. In which, as a voltage of 3.5 V was applied on ECD, a transmittance change ($\Delta T$%) of 53.1%, an optical density ($\Delta$OD) of 0.502, an intercalation charge ($Q$) of 12.9 mC/cm$^2$ and a coloration efficiency ($\eta$) of 41.6 cm$^2$/C at a wavelength of 550 nm can be achieved. These results demonstrate that Li$_2$O doping in WO$_3$ films could effectively improve the coloration and electrochromic properties of ECD devices.

**Keywords:** tungsten trioxide; lithium oxide; electrochromic device; electron beam evaporation

## 1. Introduction

With the rapid development in science and technology, people are projected to face an attendant energy crisis. In addition to the continual development of new energy sources, effective methods must be established for energy-saving systems. Electrochromic materials have been widely investigated and are expected to replace traditional energy-saving glasses [1]. The electrochromic phenomenon was first reported in 1961 by Platt, who revealed that an excited material shows a new light absorption band because of the transfer of electrons through migration or redox reactions, which results in color changes in materials. Furthermore, the loss of electrons that have absorbed considerable energy from sunlight (visible and infrared spectra) results in the material exhibiting a faded appearance (coloring-bleaching) [2]. In 1969 at the SERI laboratory, Deb discovered that a tungsten trioxide (WO$_3$) film showed electrochromic properties [3]. The properties of electrochromic films are closely related to the amount of charge storage in the film. In general, Li$^+$ is adopted as the transportation ion in the electrochromic devices (ECDs) because of its fast charge transfer rate in the film [4]. It has been revealed that increasing the amount of charges storage by doping lithium into tungsten trioxide (WO$_3$) using the sol-gel process will result in better characteristics of electrochromic devices [5–7].

In this study, electron beam evaporation technology was adopted to fabricate electrochromic films of Li$_x$W$_{1-x}$O. Various amounts of Li$_2$O were mixed into WO$_3$ to obtain Li$_x$W$_{1-x}$O powders.

Electrochromic films were deposited onto indium tin oxide (ITO) glasses using the optimized parameters. The gel polymer electrolyte containing lithium perchlorate was synthesized to be an ion storage layer. In this structure of electrochromic device, both electrolyte and electrochromic films contain lithium ions that will enhance the rate of charge transfer to improve the ECD characteristics. ECDs were fabricated and their characteristics were investigated, including the light transmittance variations ($\Delta T\%$) and cyclic voltammetry (CV) behaviors.

## 2. Experimental

### 2.1. Preparation of Materials

Various amounts of $Li_2O$ were mixed into $WO_3$ to obtain the $Li_xW_{1-x}O$ powders in accordance with the particular atomic ratio using a high-speed crushing machine for 30 s. Circular powder tablets with a diameter of 1.5 cm and a thickness of 1 cm were formed using a hot press apparatus. The samples with different atomic ratio of $WO_3$, $Li_{0.1}W_{0.9}O_{2.75}$, $Li_{0.2}W_{0.8}O_{2.5}$, and $Li_{0.3}W_{0.7}O_{2.25}$ were named as $WO_3$, LW1, LW2, and LW3 respectively, as shown in Table 1.

**Table 1.** Samples with different amounts of $Li_2O$ doping in $WO_3$.

| $Li_2O$ (wt.%) | Name | $Li_xW_{1-x}O$ |
|:---:|:---:|:---:|
| 0 | $WO_3$ | $WO_3$ |
| 0.71 | LW1 | $Li_{0.1}W_{0.9}O_{2.75}$ |
| 1.58 | LW2 | $Li_{0.2}W_{0.8}O_{2.5}$ |
| 2.68 | LW3 | $Li_{0.3}W_{0.7}O_{2.25}$ |

### 2.2. Preparation of Electrochromic Devices

In this study, electrochromic films were deposited by electron beam evaporation (ULVAC, CRTM 6000, Chigasaki, Japan). ITO glasses were chosen as substrates. The substrate was cut into size of 3 cm × 3.5 cm with a thickness of ITO film of about 250 nm and sheet resistance of about 7 $\Omega$. The ITO glass was placed above the tablet target put in a crucible. The distance between the ITO glass and the target was 20 cm. The base pressure of the chamber was 0.65 mPa. The deposition parameters of the electrochromic films are shown in Table 2.

**Table 2.** The deposition parameters of electrochromic films.

| Material | $WO_3$ LWO |
|:---:|:---:|
| Substrate | ITO/Glass (ITO: 7 $\Omega$, 250 nm) |
| Accelerating voltage (kV) | 4 |
| Deposition rate (nm/s) | 0.5 |
| Rotation speed (rpm) | 40 |
| Base pressure (mPa) | 0.65 |
| Deposition temperature (K) | R.T. |
| Film thickness (nm) | 530 |
| Oxygen pressure (mPa) | none |

ECDs were constructed with an electrochromic layer and an electronically conducting transparent substrate which is separated by an ion storage layer (gel polymer electrolyte, GPE). Figure 1 shows the schematic of the fabrication process of GPE. Lithium perchlorate ($LiClO_4$) powder was dispersed in propylene carbonate (PC) solvent to complete 1 M electrolyte. Then, 4.5 wt.% of ethyl-cellulose and 8 wt.% ethylene carbonate (EC) were added to the electrolyte under stirring to form the GPE [8]. Finally, the electrochromic thin films obtained with the optimized deposition parameters were combined with the GPE to prepare the ECD structure of glass/ITO/$Li_xW_{1-x}O$/GPE/ITO/glass.

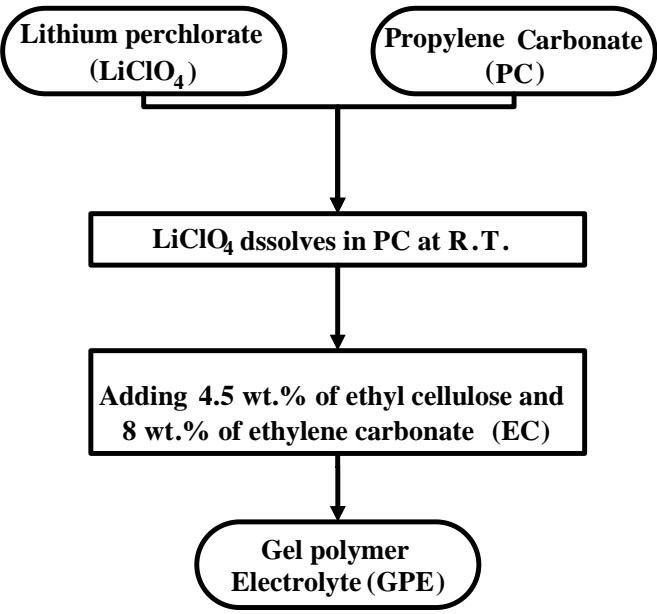

**Figure 1.** The fabrication process of the gel polymer electrolyte (GPE).

*2.3. Characterization and Analysis of Electrochromic Properties*

The crystalline structures of the electrochromic films were analyzed using X-ray diffraction (XRD, Bruker D8 Advance, Billerica, MA, USA) with Cu-K$\alpha$ radiation ($\lambda$ = 0.1542 nm). Step scans with a scan rate of 4°/min were performed in the 2$\theta$ range from 20° to 70°. The surface microstructures of the films were investigated using a scanning electron microscope (SEM, ZEISS, Auriga-39-50, Oberkochen, Germany). The properties of the electrochromic films were characterized in a two-electrode cell with an electrochemical analyzer (CHI, 6273B, Austin, TX, USA), in which, the Li$_x$W$_{1-x}$O/ITO/glass was a working electrode and a common ITO/glass was simultaneously a counter electrode and a reference electrode [9]. The atomic composition of the material was analyzed by Electron Spectroscopy for Chemical Analysis (ESCA, JEOL, JAMP-9500F, Tokyo, Japan) [10].

The cyclic voltammetry (CV) measurements of the ECDs were carried out using potential sweeps of 50 mV/s from −3.5 to +3.5 V. To calculate the inserted charge (*Q*) for the coloration states and bleaching states of films, Equation (1) was adopted for integrating between the starting and ending time of each period:

$$Q = \int_{t_1}^{t_2} I(t)\,dt\,(\mathrm{mC/cm^2}) \tag{1}$$

where *t* is scanning time and *j* is current density.

The optical density change ($\Delta$OD) is given by [11]:

$$\Delta\mathrm{OD} = \log\left(\frac{T_{bleached}}{T_{colored}}\right) \tag{2}$$

where $T_{\mathrm{bleached}}$ and $T_{\mathrm{colored}}$ represent the transmittance of the bleaching and coloring states, respectively at a wavelength of 550 nm.

Optical transmittance spectra were measured using an ultraviolet–visible–near-infrared (UV-visible-NIR) spectrophotometer (Jasco, V-570, Easton, MD, USA) in the range of 200–2500 nm wavelength. Transmittance data were taken at or near the center of each cell, depending on the size of the sample. The transmittance of the device was measured against air as a reference. The coloration efficiency ($\eta$) is defined as follows:

$$\eta = \frac{\Delta\mathrm{OD}}{Q}\,(\mathrm{cm^2/C}) \tag{3}$$

## 3. Results and Discussion

Figure 2 shows the XRD patterns of ITO, $WO_3$, LW1, LW2, and LW3 thin films. The results show that all the peaks of crystalline phases that appeared belong to the ITO substrates, it means that $WO_3$ and $Li_2O$- doped $WO_3$ films deposited on the ITO substrates all are amorphous. The reason may due that the films were deposited at room temperature, and the energy was insufficient to induce crystalline growth [12]. Figure 3 shows the surface morphologies of electrochromic films deposited on the ITO substrates. It shows that all the surfaces of the films are smooth and compact. Figure 4 shows the surface roughness of the deposited films. The roughness increases with the amount of $Li_2O$ doped in $WO_3$, which may help the ion migration process in the electrochromic films.

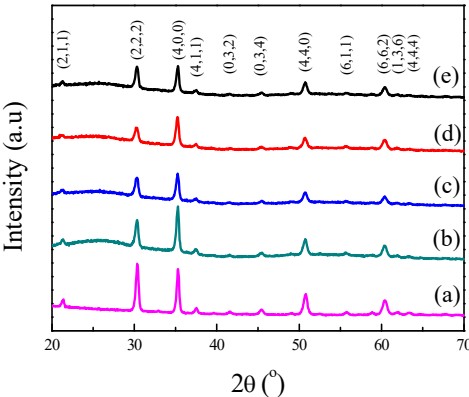

**Figure 2.** The X-ray diffraction (XRD) patterns of electrochromic films: (**a**) ITO, (**b**) $WO_3$, (**c**) LW1, (**d**) LW2, (**e**) LW3.

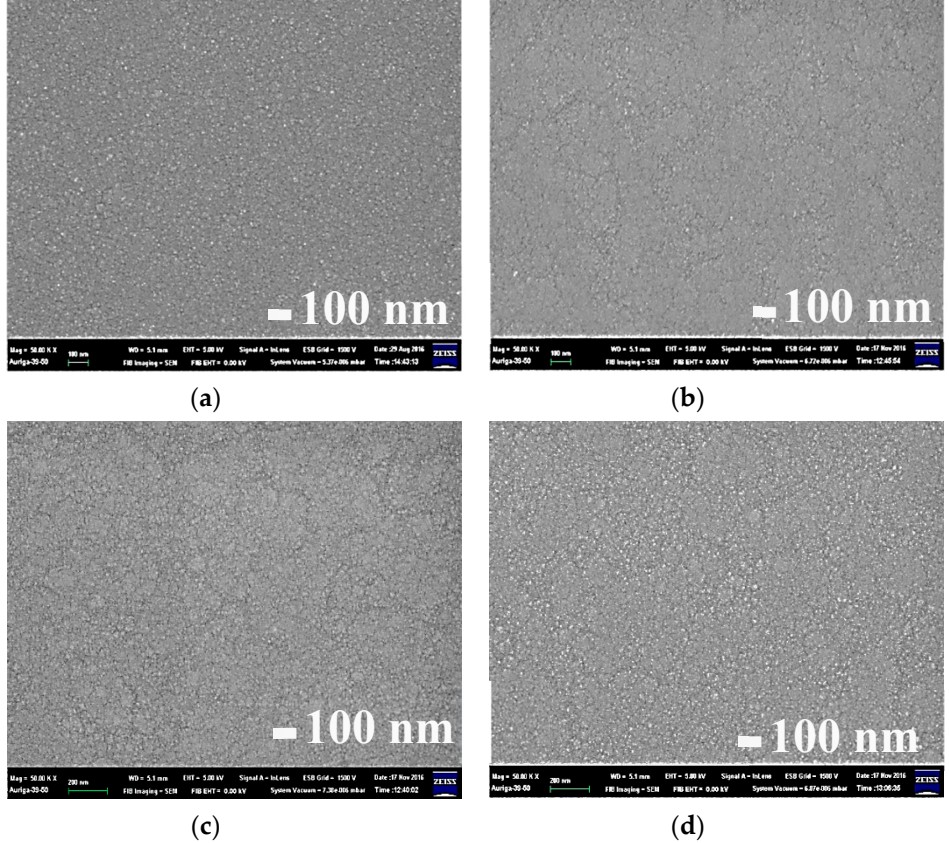

**Figure 3.** Surface morphologies of electrochromic films: (**a**) $WO_3$; (**b**) LW1; (**c**) LW2; (**d**) LW3.

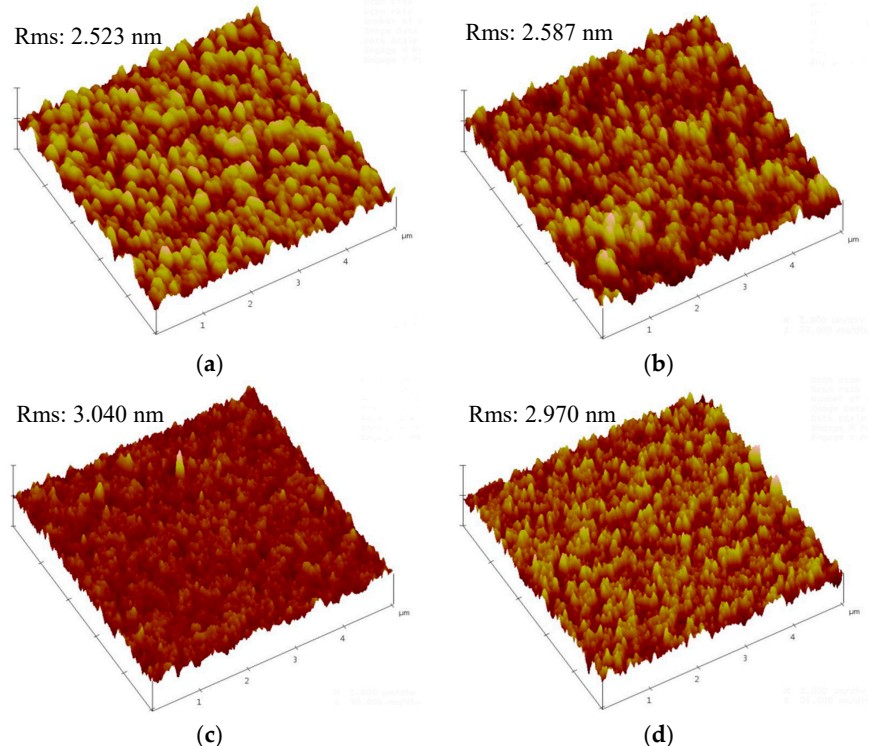

**Figure 4.** Atomic force microscopies of electrochromic films: (**a**) WO$_3$; (**b**) LW1; (**c**) LW2; (**d**) LW3.

Figure 5 shows the cyclic voltammetry characteristics of the ECDs applied with a coloring voltage of 3.5 V and bleaching voltage of −3.5 V, with a scanning loop of 0 V→−3.5 V→0 V→3.5 V→0 V and a fixed scan rate of 50 mV/s. The peak at 0 V is due that the Li$^+$ ion adsorption in ITO glass was difficult as ITO glass was used as the counter electrode and the ion will migrate into the electrochromic films on ITO glasses. In addition, the peak appeared at −3.5 V means that the W$^{4+}$ in WO$_3$ will gradually be oxidized into W$^{6+}$. On the other hand, the peak appeared at 2.5 V suggests that the W$^{6+}$ in WO$_3$ will gradually be reduced into W$^{4+}$. The two values of 2.5 and −3.5 V are the maximum operating voltages of the devices. The cyclic voltammetry area of WO$_3$ film is less than those of LWO films. The area of the cyclic voltammetry increases with the doping amount of Li$_2$O in WO$_3$, which may be caused by the increased surface roughness of the electrochromic films, as shown in Figure 4. The results indicate that a rough surface of electrochromic film will improve the Li$^+$/e$^-$ intercalation/deintercalation reaction during the CV process. The ECD with LW2 film exhibits the maximum cyclic voltammetry area, as shown in Figure 5. However, the cyclic voltammetry area of ECD with LW3 film decreases due to the excessive doping amount of Li$_2$O in WO$_3$. It was pointed out that the electrical conductivity of WO$_3$ can be increased by doping of Li$^+$ [13]. Whereas, excessive doping amount of ions will decrease the carrier mobility, which results in the decreased electrical conductivity and also the cyclic voltammetry area of ECD [14].

The transmittance spectra of ECDs with different doping amounts of Li$_2$O are shown in Figure 6. The spectra were taken after ECDs were applied by a bleaching voltage of 2.5 V and a coloring voltage of −3.5 V for 12 s, respectively. Figures 7 and 8 show the transmittance variations (Δ$T$%) and optical density variations (ΔOD) as a function of ECD with different doping amounts of Li$_2$O. Δ$T$% is given as $T_{bleached} - T_{colored}$, and ΔOD is given as ΔOD = log($T_{bleached}/T_{colored}$) at 550 nm in bleached and colored states, respectively. The results show that the optical properties of ECDs are dependent on different doping amount of Li$_2$O; both Δ$T$% and ΔOD increased significantly with an increase in the doping amount of Li$_2$O. The reason for this may be due to the increased surface roughness of films with increased doping amount of Li$_2$O, which may improve the Li$^+$/e$^-$ intercalation/deintercalation reaction during the coloring/bleaching process. The rough surfaces of Li$_2$O doped WO$_3$ electrochromic

films may help the ion migration process. The transmittance spectra of ECD without $Li_2O$ doping in $WO_3$ exhibit the lowest $\Delta T\%$ and $\Delta OD$. Meanwhile, ECD with LW2 film showed the best transmittance variation ($\Delta T\%$) of 53.1% and $\Delta OD$ of 0.502 at 550 nm wavelength ($\lambda$), and were effective in filtering visible and near-infrared light, i.e., it inhibited the passing of radiated heat. However, when $WO_3$ was doped with excessive amount of $Li_2O$, $\Delta T\%$ and $\Delta OD$ decreased. According to Stafstrom et al., when $\Delta OD$ is above 0.3, it is very suitable for monitors and smart glass [15].

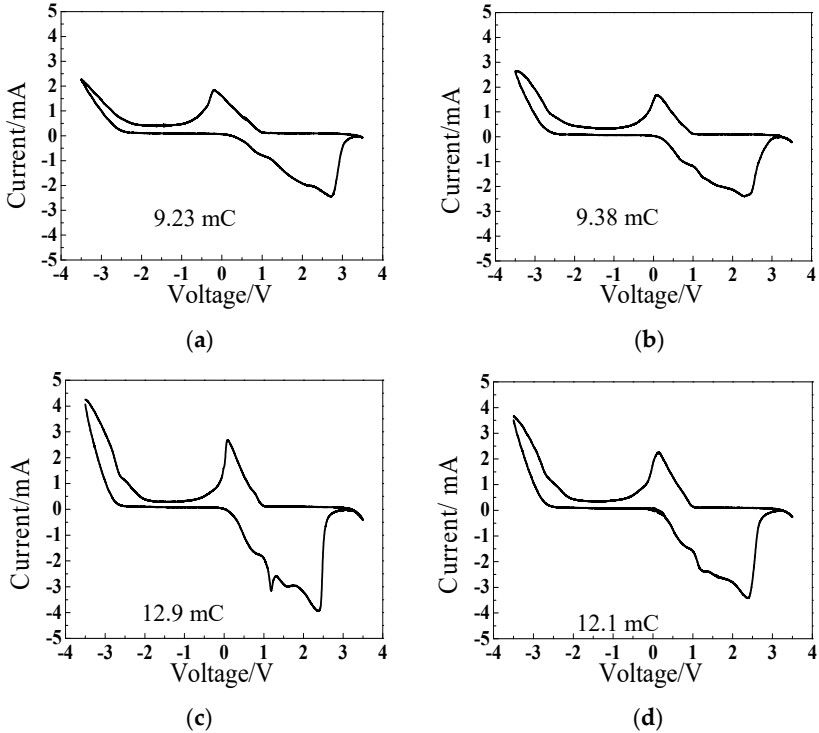

**Figure 5.** Cyclic voltammetry of electrochromic devices (ECDs): (**a**) $WO_3$; (**b**) LW1; (**c**) LW2; (**d**) LW3.

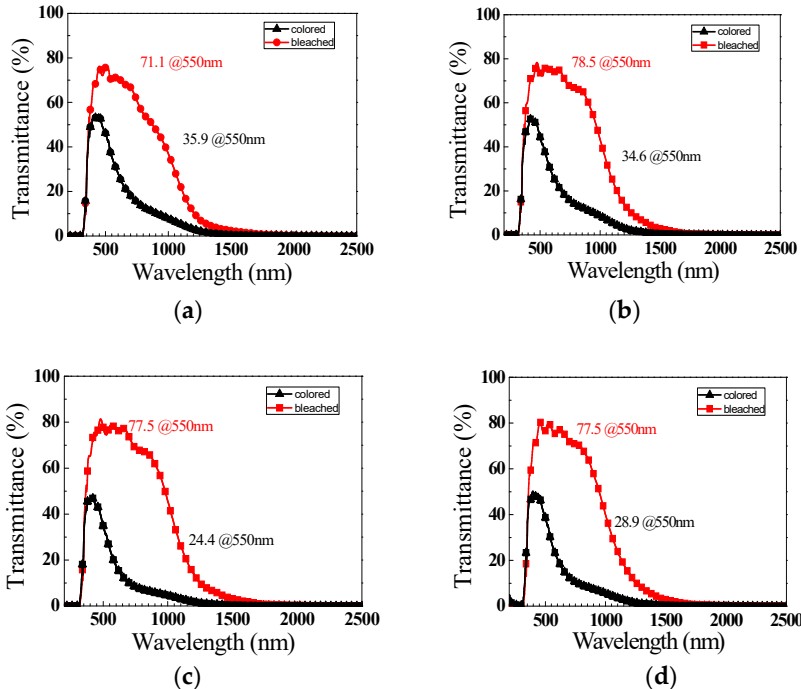

**Figure 6.** *Cont.*

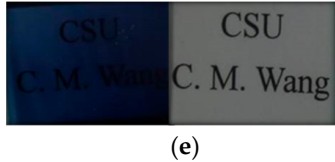

(**e**)

**Figure 6.** Transmittance spectra of ECDs: (**a**) WO$_3$; (**b**) LW1; (**c**) LW2; (**d**) LW3; (**e**) The photographs of the LW2 in coloring and bleaching state.

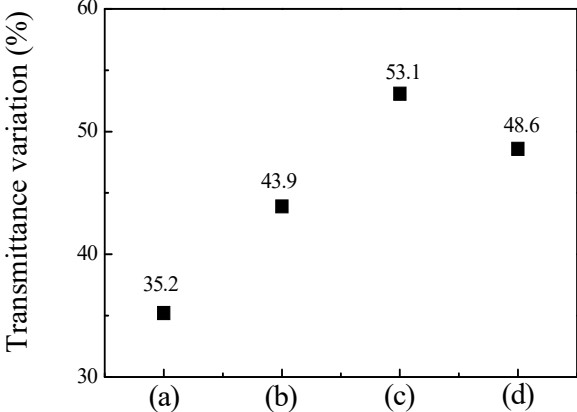

**Figure 7.** Transmittance variation ($T_{\text{bleached}} - T_{\text{colored}}$) of ECDs: (**a**) WO$_3$; (**b**) LW1; (**c**) LW2; (**d**) LW3.

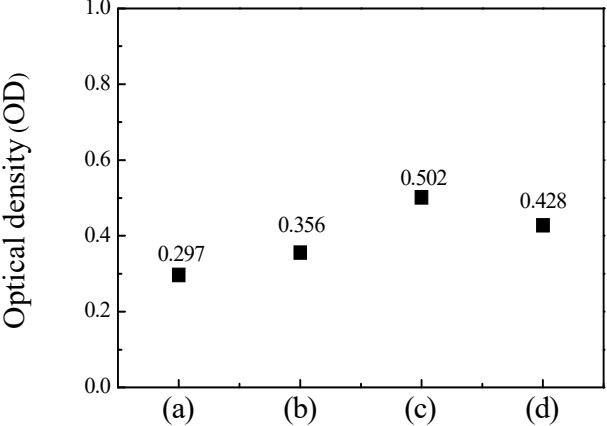

**Figure 8.** Optical density of ECDs: (**a**) WO$_3$; (**b**) LW1; (**c**) LW2; (**d**) LW3.

Coloration efficiency is one of the most important parameters for evaluating electrochromic materials and is defined as the change in optical density per unit of charge inserted/extracted into/from the electrochromic material at a given wavelength [16]. Figure 9 shows that the coloration efficiency increases with increasing doping amount of Li$_2$O in WO$_3$; the LW2 film exhibits the best coloration efficiency of 41.6 cm$^2$/C. However, a decrease in the coloration efficiency is caused by the higher doping amount of Li$_2$O, which cause the film to become denser. Details of the characteristics of the electrochromic thin films are shown in Table 3. Finally, the elementary composition analysis was carried out using ESCA for LW2 and WO$_3$. The results show that the LW2 film was analyzed to Li of 4.9 at.%, W of 22.7 at.%, and O of 72.4 at.%, and the chemical formula was calculated to be Li$_{0.18}$W$_{0.82}$O$_{2.6}$, as shown in Table 4.

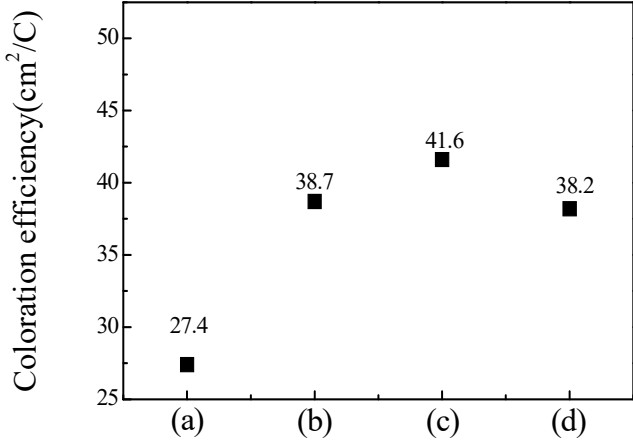

**Figure 9.** The coloration efficiency of ECDs: (**a**) WO$_3$; (**b**) LW1; (**c**) LW2; (**d**) LW3.

**Table 3.** The properties of ECDs ($\lambda$ = 550 nm).

| Film | Bleaching Transmittance ($T_b$, %) | Coloring Transmittance ($T_c$, %) | Transmittance Change ($\Delta T$ %) | Intercalation Charge ($Q$, mC/cm$^2$) | Optical Density ($\Delta$OD) | Coloration Efficiency ($\eta$, cm$^2$/C) |
|---|---|---|---|---|---|---|
| WO$_3$ | 71.1 | 35.9 | 35.2 | 10.8 | 0.297 | 27.4 |
| LW1 | 78.5 | 34.6 | 43.9 | 9.2 | 0.356 | 38.7 |
| LW2 | 77.5 | 24.4 | 53.1 | 12.1 | 0.502 | 41.6 |
| LW3 | 77.5 | 28.9 | 48.6 | 11.2 | 0.428 | 38.2 |

**Table 4.** Chemical analysis of WO$_3$ and LW2 films.

| Name | Li (at.%) | W (at.%) | O (at.%) | Li$_x$W$_{1-x}$O |
|---|---|---|---|---|
| WO$_3$ | 0 | 25.6 | 74.4 | WO$_{2.9}$ |
| LW2 | 4.9 | 22.7 | 72.4 | Li$_{0.18}$W$_{0.82}$O$_{2.6}$ |

## 4. Conclusions

In this study, electrochromic films were deposited on ITO substrates at R.T. using un-doped and Li$_2$O-doped WO$_3$. The electrochromic films exhibited amorphous structures with the smooth and dense surfaces, as revealed by the SEM analysis. Cyclic voltammetry analysis showed that the hysteresis area of cyclic voltammetry after Li$_2$O doping in WO$_3$ increased. In addition, the charge/discharge amount also increased. However, the transportation ability of ions decreased when the dopant amount exceeded a certain atomic percentage; this made the ECD characteristics worse. The LW2 film had the best ECD characteristics at 550 nm: the $\Delta T$% was 53.1%, the $\Delta$OD was 0.502, the $Q$ value was 12.9 mC/cm$^2$, and the $\eta$ was 41.6 cm$^2$/C. These results demonstrate that Li$_2$O doping can effectively improve the coloration and electrochromic properties of WO$_3$ electrochromic films.

**Author Contributions:** J.-Y.C. and W.-N.W. performed the experiments and wrote the paper. C.-Y.W. and J.-M.L. helped with the experiments. Y.-C.C. and C.-M.W. provided guidance and helped in manuscript preparation.

**Funding:** This study was supported by the Ministry of Science and Technology of the Republic of China, Taiwan (No. MOST 107-2221-E-230-006).

**Conflicts of Interest:** The authors declare no conflict of interest.

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
