# Peer review of "Electrochromic Properties of Lithium-Doped Tungsten Oxide Prepared by Electron Beam Evaporation"

_coatings, doi:10.3390/coatings9030191_

Reviewer 1 Report

The authors studied the electrochromic properties of lithium doped tungsten oxide films prepared by e-beam evaporation. The topic of the paper is suitable for Coatings. However, there are some unclear points in the manuscript and the discussions seems to be insufficient. The reviewer considers that it is necessary to make some major corrections of the paper before publication. Please consider the following comments.

1. Novelty.

The novelty of this study is not clearly described in the manuscript.

The effects of Li doping on WO3 film have already been studied by some previous research papers. The difference of this study and the previous paper should be clearly described in the introduction. The new findings obtained by this study should be clearly described in the manuscript. 

2. XRD.

The reviewer supposes that the ITO films on glass substrates are crystalline. Why XRD peaks due to ITO were not observed in Fig. 2?

3. Surface morphology.

It seems difficult to distinguish the surface morphology of the films only from the surface SEM images shown in Fig. 3. The reviewer considers that it is necessary to add cross-sectional SEM images or AFM images of the films to clarify the difference of their surface morphology.

4. Intercalation charge.

It is described that “The area of the cyclic voltammetry increases with the doping amount of Li2O in WO3” (page 5, lines 1 and 2). However, Table 3 shows that the intercalation charge of LW1 film is smaller than that of WO3 film, and the intercalation charges of WO3 and LW3 are nearly the same. Are there any comments on this discrepancy? 

It is described that “However, the cyclic voltammetry area of ECD with LW3 film decreases due to the excess doping amount of Li2O in WO3,” (page 5, lines 5 and 6). It seems necessary to explain the reason why the excess doping of Li2O induced the decrease of the intercalation charge.

5. Transmittance measurement.

Did the authors measure the transmittance spectra shown in Fig. 5 after applying constant voltages to the ECDs? If so, the applied voltages and applied times should be added in the manuscript.  

6. Density.

It is described that “However, a decrease of the coloration efficiency is caused by the higher doping amount of Li2O, which let the film become denser. (page 7, lines 5 and 6)” Are there any experimental evidence or appropriate reference papers that indicate the density of WO3 films increased by Li2O doping? 

The reviewer agrees that dense films prevent ion diffusion and the amount of intercalated charge decreases with increasing film density. However the effect of the film density on the coloration efficiency is not clear because it is expected that both of the intercalation charge and optical density change decrease with increasing film density. 

7. Film thickness.

The film thickness described in the abstract is about 540 nm, however, that shown in Table 2 is 530 nm. The discrepancy should be corrected.

How did the authors measure the film thickness? The experimental method to evaluate the film thickness should be added in the manuscript because the film thickness affects the EC properties of the films.

8. Film composition.

The composition of the LW2 is described as Li0.16W0.17O2.5 in the text (page 7, line9), however, that shown in Table 4 is Li0.18W0.82O2.6. The discrepancy should be corrected. 

9. English.

There are some inadequate English expressions in the manuscript. It seems better to get English proofreading. 

Author Response

Comments and Suggestions for Authors

The authors studied the electrochromic properties of lithium doped tungsten oxide films prepared by e-beam evaporation. The topic of the paper is suitable for Coatings. However, there are some unclear points in the manuscript and the discussions seems to be insufficient. The reviewer considers that it is necessary to make some major corrections of the paper before publication. Please consider the following comments.

1. Novelty.

The novelty of this study is not clearly described in the manuscript.

The effects of Li doping on WO3 film have already been studied by some previous research papers. The difference of this study and the previous paper should be clearly described in the introduction. The new findings obtained by this study should be clearly described in the manuscript.

Ans: Thanks for reviewer’s comment. The more descriptions are added in page 1, lines 36~39 and page 2, lines 5~7. (The properties of electrochromic films are closely related to the amount of charge storage in the film. In general, Li+ is adopted as the transportation ion in the electrochromic devices (ECDs) because of its fast charge transfer rate in the film [4]. It has been revealed that increasing the amount of charges storage by doping lithium into tungsten trioxide (WO3) using the sol-gel process will result in better characteristics of electrochromic devices [5-7].

The gel polymer electrolyte containing lithium perchlorate was synthesized to be an ion storage layer. In this structure of electrochromic device, both electrolyte and electrochromic films contain lithium ions that will enhance the rate of charge transfer to improve the ECD characteristics.)

2. XRD.

The reviewer supposes that the ITO films on glass substrates are crystalline. Why XRD peaks due to ITO were not observed in Fig. 2?

Ans: Thanks for reviewer’s comment. The more descriptions are added and corrected in page 4, lines 29~31. (Figure 2 shows the XRD patterns of ITO, WO3, LW1, LW2, and LW3 thin films. The results show that all the peaks of crystalline phases appeared belong to the ITO substrates, it means that WO3 and Li2O- doped WO3 films deposited on the ITO substrates all are amorphous.)

 Figure 2. The X-ray diffraction (XRD) patterns of electrochromic films: (a) ITO, (b)WO3, (c)LW1, (d)LW2, (e)LW3.

3. Surface morphology.

It seems difficult to distinguish the surface morphology of the films only from the surface SEM images shown in Fig. 3. The reviewer considers that it is necessary to add cross-sectional SEM images or AFM images of the films to clarify the difference of their surface morphology.

Ans: Thanks for reviewer’s comment. The more descriptions are added in page4, lines34~ 36 and page6, lines1~ 23. (It shows that all the surfaces of the films are smooth and compact. Figure 4 shows the surface roughness of the deposited films. The roughness increases with the amount of Li2O doped in WO3, which may help the ion migration process in the electrochromic films.)

Figure 4. Atomic force microscopies of electrochromic films: (a)WO3; (b)LW1; (c)LW2; (d)LW3.

4. Intercalation charge.

It is described that “The area of the cyclic voltammetry increases with the doping amount of Li2O in WO3” (page 5, lines 1 and 2). However, Table 3 shows that the intercalation charge of LW1 film is smaller than that of WO3 film, and the intercalation charges of WO3 and LW3 are nearly the same. Are there any comments on this discrepancy? 

It is described that “However, the cyclic voltammetry area of ECD with LW3 film decreases due to the excess doping amount of Li2O in WO3,” (page 5, lines 5 and 6). It seems necessary to explain the reason why the excess doping of Li2O induced the decrease of the intercalation charge.

Ans: Thanks for reviewer’s comment. The more descriptions are added in page 6, lines 38 and page 7, lines 1~ 3. (It was pointed out that the electrical conductivity of WO3 can be increased by doping of Li+ [13]. Whereas, excessive doping amount of ions will decrease the carrier mobility, which results in the decreased electrical conductivity and also the cyclic voltammetry area of ECD [14].)

5. Transmittance measurement.

Did the authors measure the transmittance spectra shown in Fig. 5 after applying constant voltages to the ECDs? If so, the applied voltages and applied times should be added in the manuscript.  

Ans: Thanks for reviewer’s comment. The more descriptions for the UV-Vis spectra of this study are added in page 7, lines 30~ 31. (The spectra were taken after ECDs were applied by a bleaching voltage of 2.5 V and a coloring voltage of -3.5 V for 12 s, respectively.)

6. Density.

It is described that “However, a decrease of the coloration efficiency is caused by the higher doping amount of Li2O, which let the film become denser. (page 7, lines 5 and 6)” Are there any experimental evidence or appropriate reference papers that indicate the density of WO3 films increased by Li2O doping?

The reviewer agrees that dense films prevent ion diffusion and the amount of intercalated charge decreases with increasing film density. However the effect of the film density on the coloration efficiency is not clear because it is expected that both of the intercalation charge and optical density change decrease with increasing film density.

Ans: Thanks for reviewer’s comment. The more descriptions are added in page 6, lines 38 and page 7, lines 1~ 3. (It was pointed out that the electrical conductivity of WO3 can be increased by doping of Li+ [13]. Whereas, excessive doping amount of ions will decrease the carrier mobility, which results in the decreased electrical conductivity and also the cyclic voltammetry area of ECD [14].)

7. Film thickness.

The film thickness described in the abstract is about 540 nm, however, that shown in Table 2 is 530 nm. The discrepancy should be corrected.

How did the authors measure the film thickness? The experimental method to evaluate the film thickness should be added in the manuscript because the film thickness affects the EC properties of the films.

Ans: Thanks for reviewer’s comment. The film thickness in the abstract is corrected, and the film thickness was measured by SEM and the optimal thickness was studied by our previous research but not present.

Table1. The thickness of electrochromic films by our previous research.

Film

thickness

Bleaching   transmittance (Tb. %)

Coloring   transmittance (Tc. %)

transmittance change

 (ΔT   %)

intercalation charge

(Q,   mC/cm2)

optical density

OD)

coloration efficiency (η, cm2/C)

440

69.4

16.6

52.8

16.3

0.621

38.2

530

77.5

24.4

53.1

12.1

0.502

41.6

620

69.8

12.5

57.3

19.1

0.747

39.1

8. Film composition.

The composition of the LW2 is described as Li0.16W0.17O2.5 in the text (page 7, line9), however, that shown in Table 4 is Li0.18W0.82O2.6. The discrepancy should be corrected. 

Ans: Thanks for reviewer’s comment. This mistake is corrected in manuscript.

9. English.

There are some inadequate English expressions in the manuscript. It seems better to get English proofreading. 

Ans: Thanks for reviewer’s comment. The English are corrected by MDPI English Editing.

Reviewer 2 Report

The article presents to my knowledge for the first time in-situ Li doping of WO3 by means of electron beam evaporation. Which is an interesting task as it e. g. might could replace wet lithiating procedures. The publication however exhibits some minor and major inaccuracies:

·  Reference [4] (Inaba, et al.), does not deal with deal with Lithium (co-) doping as stated in the introduction of the presented paper. It simply deals with intercalation after WO3 deposition.

·  Units must be SI-Units, i. e. torr must be converted to mPa, K instead of C°

·  Unit of sheet resistance is simply Ω, not Ω□ (Page 2)

·  Page2: I can hardly understand how your device is constructed/working. Please clarify. Common device setups are:

Glass/ITO/WO3/electrolyte/ion storage layer/ITO/Glass, i. e. the exhibit two layers able to intercalate Lithium, wich is moved from one layer to the other.

In your setup the ion storage layer is missing:

Glass/ITO/LixW1-xO/GPE/ITO/Glass.

Furthermore your description of the GPE is confusing “…which is separated by an ion storage layer (gel polymer electrolyte)..” Is the GPE an ion storage layer OR an electrolyte? Please clarify. If the GPE is simply an electrolyte, please comment why your setup is nevertheless working. To my opinion, ITO has only some very little irreversible intercalation (ion storage) properties. The device might only work some cycles.

·   ..dsscolves à dissolves (Page 3)

·   ..In Figure 3(a), the surface of WO3 film is compact and smoother than those of Li2O doped WO3, as shown in Figure 3(b)-(d). à I can’t see enhanced roughness for b) to d). Are there any e. g. AMF or other results supporting this opinion?

·   …The cyclic voltammetry (CV) measurements of the electrochromic devices were carried out using potential sweeps of 50 mV/s from -3.5 to +3.5 V. (Page 3)

The CV-Diagrams also show a way back form +3.5 V to -3.5 V. Also within the CV diagrams, the moving direction is not indicated. Please indicate, e. g with arrows. Furthermore one would be interested in the initial coloration state of the
LixW1-xO layer directly after deposition: Colored due to Li doping/intercalation? Bleached or inbetween? At which voltage the CV was started? How many CV cycles were conducted? Is there a different behavior with increasing cycle numbers? Are there any degradation effects? Which cycle (number) is shown in the CV-Diagram? Is there any special lithiating (i. e. lithium intercalation) procedure in the beginning? Is the determined intercalation charge Q the same as the deinercalated charge? Three “peaks” can be seen at approximately +2.5 V, 0 V and -3.0 V. What is the nature of these three peaks?

·  Please comment whether the incorporated (during deposition)Li acts as (electrochromic inactive) doping or is it electrochromic active? or something inbetween?

·  Table 3. Exhibits major errors. The calculation of the coloration efficiency is incorrect. Numbers I calculate for WO3, LW1, LW2, LW3 are 27.5, 41.8, 38.9, 38.6 cm2/C. I. e. samples or numbers were mixed or something different went wrong. Therefore, also Figure 8 and any conclusions drawn from table or figure are questionable.

·  A comparison/discussion of the obtained findings with respect to literature findings e. g. from Ref. 4­-7 is missing and should be done.

Author Response

 Comments and Suggestions for Authors

The article presents to my knowledge for the first time in-situ Li doping of WO3 by means of electron beam evaporation. Which is an interesting task as it e. g. might could replace wet lithiating procedures. The publication however exhibits some minor and major inaccuracies:

·Reference [4] (Inaba, et al.), does not deal with deal with Lithium (co-) doping as stated in the introduction of the presented paper. It simply deals with intercalation after WO3deposition.

Ans: Thanks for reviewer’s comment. This mistake is corrected in manuscript.

·Units must be SI-Units, i. e. torr must be converted to mPa, K instead of C°

·Unit of sheet resistance is simply Ω, not Ω□ (Page 2)

Ans: Thanks for reviewer’s comment. This mistake is corrected in manuscript.

Page2: I can hardly understand how your device is constructed/working. Please clarify. Common device setups are:

Glass/ITO/WO3/electrolyte/ion storage layer/ITO/Glass, i. e. the exhibit two layers able to intercalate Lithium, wich is moved from one layer to the other.

In your setup the ion storage layer is missing:

Glass/ITO/LixW1-xO/GPE/ITO/Glass.

Furthermore your description of the GPE is confusing “…which is separated by an ion storage layer (gel polymer electrolyte)..” Is the GPE an ion storage layer OR an electrolyte? Please clarify. If the GPE is simply an electrolyte, please comment why your setup is nevertheless working. To my opinion, ITO has only some very little irreversible intercalation (ion storage) properties. The device might only work some cycles.

.dsscolves à dissolves (Page 3)

Ans: Thanks for reviewer’s comment. The more descriptions are added page 1, lines 36~39 and page 2, lines 5~7. (The properties of electrochromic films are closely related to the amount of charge storage in the film. In general, Li+ is adopted as the transportation ion in the electrochromic devices (ECDs) because of its fast charge transfer rate in the film [4]. It has been revealed that increasing the amount of charges storage by doping lithium into tungsten trioxide (WO3) using the sol-gel process will result in better characteristics of electrochromic devices [5-7].

The gel polymer electrolyte containing lithium perchlorate was synthesized to be an ion storage layer. In this structure of electrochromic device, both electrolyte and electrochromic films contain lithium ions that will enhance the rate of charge transfer to improve the ECD characteristics.)

In Figure 3(a), the surface of WO3 film is compact and smoother than those of Li2O doped WO3, as shown in Figure 3(b)-(d). à I can’t see enhanced roughness for b) to d). Are there any e. g. AMF or other results supporting this opinion?

Ans: Thanks for reviewer’s comment. The more descriptions are added in page4, lines34~ 36 and page6, lines1~ 23. (It shows that all the surfaces of the films are smooth and compact. Figure 4 shows the surface roughness of the deposited films. The roughness increases with the amount of Li2O doped in WO3, which may help the ion migration process in the electrochromic films.)

Figure 4. Atomic force microscopies of electrochromic films: (a)WO3; (b)LW1; (c)LW2; (d)LW3.

The cyclic voltammetry (CV) measurements of the electrochromic devices were carried out using potential sweeps of 50 mV/s from -3.5 to +3.5 V. (Page 3)
The CV-Diagrams also show a way back form +3.5 V to -3.5 V. Also within the CV diagrams, the moving direction is not indicated. Please indicate, e. g with arrows. Furthermore one would be interested in the initial coloration state of the LixW1-xO layer directly after deposition: Colored due to Li doping/intercalation? Bleached or inbetween? At which voltage the CV was started? How many CV cycles were conducted? Is there a different behavior with increasing cycle numbers? Are there any degradation effects? Which cycle (number) is shown in the CV-Diagram? Is there any special lithiating (i. e. lithium intercalation) procedure in the beginning? Is the determined intercalation charge Q the same as the deinercalated charge? Three “peaks” can be seen at approximately +2.5 V, 0 V and -3.0 V. What is the nature of these three peaks?

Ans: Thanks for reviewer’s comment. The more descriptions are added in page 6, lines 25~ 32.( Figure 5 shows the cyclic voltammetry characteristics of the ECDs applied with a coloring voltage of 3.5 V and bleaching voltage of -3.5 V, with a scanning loop of 0 V→-3.5 V→0 V→3.5 V→0 V and a fixed scan rate of 50 mV/s. The peak at 0 V is due that the Li+ ion adsorption in ITO glass was difficult as ITO glass was used as the counter electrode and the ion will migrate into the electrochromic films on ITO glasses. In addition, the peak appeared at -3.5 V means that the W4+ in WO3 will gradually be oxidized into W6+. On the other hand, the peak appeared at 2.5V suggests that the W6+ in WO3 will gradually be reduced into W4+. The two values of 2.5 V and -3.5 V are the maximum operating voltages of the devices.)

Please comment whether the incorporated (during deposition)Li acts as (electrochromic inactive) doping or is it electrochromic active? or something inbetween?

Ans: Thanks for reviewer’s comment. The ECDs were increased the electrical conductivity and ion store by doping Li.

Table 3.Exhibits major errors. The calculation of the coloration efficiency is incorrect. Numbers I calculate for WO3, LW1, LW2, LW3 are 27.5, 41.8, 38.9, 38.6 cm2/C. I. e. samples or numbers were mixed or something different went wrong. Therefore, also Figure 8 and any conclusions drawn from table or figure are questionable.

Ans: Thanks for reviewer’s comment. This mistake is corrected in manuscript.

A comparison/discussion of the obtained findings with respect to literature findings e. g. from Ref. 4­-7 is missing and should be done.

Ans: Thanks for reviewer’s comment. This mistake is corrected in manuscript.

Reviewer 3 Report

The work reports the electrochromic properties of lithium doped tungsten oxide films prepared by electron beam evaporation. The reviewer considers that the article is very incomplete and that the conclusions are not fully addressed by the results, apart from that the work did not bring any novelty to the field. Thus, the article is not suitable for publication in Coatings. However, the reviewer suggests that the characterization of materials can be improved. The images of morphology are weak in information and can be optimized. Furthermore, the characterization of electrochromic properties can be improved, determining the response times and the electrochemical stability. Photographs of the assembled devices should be present and the UV-Vis spectra revised. The English language should also be revised.

Author Response

Comments and Suggestions for Authors

The work reports the electrochromic properties of lithium doped tungsten oxide films prepared by electron beam evaporation. The reviewer considers that the article is very incomplete and that the conclusions are not fully addressed by the results, apart from that the work did not bring any novelty to the field. Thus, the article is not suitable for publication in Coatings. However, the reviewer suggests that the characterization of materials can be improved. The images of morphology are weak in information and can be optimized. Furthermore, the characterization of electrochromic properties can be improved, determining the response times and the electrochemical stability. Photographs of the assembled devices should be present and the UV-Vis spectra revised. The English language should also be revised.

1.    In “Introduction” part:

Ans: Thanks for reviewer’s comment. The more descriptions for the motivation of this study are added in page 1, lines 36~39 and page 2, lines 5~7. (The properties of electrochromic films are closely related to the amount of charge storage in the film. In general, Li+ is adopted as the transportation ion in the electrochromic devices (ECDs) because of its fast charge transfer rate in the film [4]. It has been revealed that increasing the amount of charges storage by doping lithium into tungsten trioxide (WO3) using the sol-gel process will result in better characteristics of electrochromic devices [5-7].

The gel polymer electrolyte containing lithium perchlorate was synthesized to be an ion storage layer. In this structure of electrochromic device, both electrolyte and electrochromic films contain lithium ions that will enhance the rate of charge transfer to improve the ECD characteristics.)

 2. In “Results and discussion” part:
2-1. Physical analysis

Ans: Thanks for reviewer’s comment. The more descriptions for the XRD of this study are added and corrected in page 4, lines 29~31. (Figure 2 shows the XRD patterns of ITO, WO3, LW1, LW2, and LW3 thin films. The results show that all the peaks of crystalline phases appeared belong to the ITO substrates, it means that WO3 and Li2O- doped WO3 films deposited on the ITO substrates all are amorphous.)

Figure 2. The X-ray diffraction (XRD) patterns of electrochromic films: (a) ITO, (b)WO3, (c)LW1, (d)LW2, (e)LW3.

The more descriptions for the AFM of this study are added in page 4, lines 34~ 36 and page 6, lines 1~ 23.

(It shows that all the surfaces of the films are smooth and compact. Figure 4 shows the surface roughness of the deposited films. The roughness increases with the amount of Li2O doped in WO3, which may help the ion migration process in the electrochromic films.)

Figure 4. Atomic force microscopies of electrochromic films: (a)WO3; (b)LW1; (c)LW2; (d)LW3.

2-2. Electrical analysis

Ans: Thanks for reviewer’s comment. The more descriptions for the cyclic voltammetry of this study are added in page 6, lines 25~ 32.( Figure 5 shows the cyclic voltammetry characteristics of the ECDs applied with a coloring voltage of 3.5 V and bleaching voltage of -3.5 V, with a scanning loop of 0 V→-3.5 V→0 V→3.5 V→0 V and a fixed scan rate of 50 mV/s. The peak at 0 V is due that the Li+ ion adsorption in ITO glass was difficult as ITO glass was used as the counter electrode and the ion will migrate into the electrochromic films on ITO glasses. In addition, the peak appeared at -3.5 V means that the W4+ in WO3 will gradually be oxidized into W6+. On the other hand, the peak appeared at 2.5V suggests that the W6+ in WO3 will gradually be reduced into W4+. The two values of 2.5 V and -3.5 V are the maximum operating voltages of the devices.)

The more descriptions for the cyclic voltammetry of this study are added in page 6, lines 38 and page 7, lines 1~ 3. (It was pointed out that the electrical conductivity of WO3 can be increased by doping of Li+ [13]. Whereas, excessive doping amount of ions will decrease the carrier mobility, which results in the decreased electrical conductivity and also the cyclic voltammetry area of ECD [14].)

Ans: Thanks for reviewer’s comment. The more descriptions for the UV-Vis spectra of this study are added in page 7, lines 30~ 31. (The spectra were taken after ECDs were applied by a bleaching voltage of 2.5 V and a coloring voltage of -3.5 V for 12 s, respectively.)

3. Other

Ans: Thanks for reviewer’s comment. a. The English are corrected by MDPI English Editing.

b. The photographs of the ECD in bleaching and coloring state are showed below:

Round  2

Reviewer 1 Report

The authors revised their paper satisfactorily. The reviewer considers that the paper is acceptable for publication.

Reviewer 3 Report

Although I consider that the authors's responses could be more convinced, considering their effort, I suggest the publication of the manuscript. However, the photographs of the device should be presented in the manuscript and not only in the cover letter!